# Unlocking the Medicinal Mysteries: Preventing Lacunar Stroke with Drug Repurposing

**DOI:** 10.3390/biomedicines12010017

**Published:** 2023-12-20

**Authors:** Linjing Zhang, Fan Wang, Kailin Xia, Zhou Yu, Yu Fu, Tao Huang, Dongsheng Fan

**Affiliations:** 1Department of Neurology, Peking University Third Hospital, Beijing 100191, China; zhanglinjing@bjmu.edu.cn (L.Z.); yifan.1103@163.com (F.W.); kllook@pku.edu.cn (K.X.); yuzhou_1995@126.com (Z.Y.); lilac_fu@126.com (Y.F.); 2Department of Epidemiology and Biostatistics, School of Public Health, Peking University, Beijing 100871, China; 3Center for Intelligent Public Health, Institute for Artificial Intelligence, Peking University, Beijing 100871, China; 4Beijing Key Laboratory of Biomarker and Translational Research in Neurodegenerative Diseases, Beijing 100191, China; 5Key Laboratory for Neuroscience, National Health Commission/Ministry of Education, Peking University, Beijing 100871, China

**Keywords:** stroke, lacunar, mendelian randomization analysis, drug repurposing, antihypertensive agent, hypolipidemic agents, hypoglycemic agents, aging, healthy, precision medicine, genetic association studies, preventive medicine

## Abstract

Currently, only the general control of the risk factors is known to prevent lacunar cerebral infarction, but it is unknown which type of medication for controlling the risk factors has a causal relationship with reducing the risk of lacunar infarction. To unlock this medical mystery, drug-target Mendelian randomization analysis was applied to estimate the effect of common antihypertensive agents, hypolipidemic agents, and hypoglycemic agents on lacunar stroke. Lacunar stroke data for the transethnic analysis were derived from meta-analyses comprising 7338 cases and 254,798 controls. We have confirmed that genetic variants mimicking calcium channel blockers were found to most stably prevent lacunar stroke. The genetic variants at or near *HMGCR*, *NPC1L1*, and *APOC3* were predicted to decrease lacunar stroke incidence in drug-target MR analysis. These variants mimic the effects of statins, ezetimibe, and antisense anti-*apoC3* agents, respectively. Genetically proxied *GLP1R* agonism had a marginal effect on lacunar stroke, while a genetically proxied improvement in overall glycemic control was associated with reduced lacunar stroke risk. Here, we show that certain categories of drugs currently used in clinical practice can more effectively reduce the risk of stroke. Repurposing several drugs with well-established safety and low costs for lacunar stroke prevention should be given high priority when doctors are making decisions in clinical practice. This may contribute to healthier brain aging.

## 1. Introduction

Lacunar stroke is a small subcortical infarct that arises from ischemia in the territory of the deep perforating arteries of the brain [1]. These arteries, also known as lenticulostriate arteries, supply blood to the brain’s deep structures, including the basal ganglia, thalamus, internal capsule, and white matter. Lacunar stroke accounts for one-quarter of the overall number of ischemic strokes. Despite their small size, they can have significant impacts on a person’s health and quality of life. They can lead to long-term intellectual and physical disabilities, including difficulties with movement, speech, and cognitive functions. The exact cause of lacunar strokes is not fully understood, but they are often associated with conditions that affect the health of blood vessels, such as hypertension, diabetes, and high cholesterol. These conditions can lead to the hardening and narrowing of the small blood vessels in the brain, reducing blood flow and increasing the risk of a lacunar stroke. Preventive treatments are generally aimed at controlling these three risk factors. However, it is not yet known which specific medication for controlling these three risk factors can have a causal relationship with reducing the risk of lacunar infarction. In other words, the exact relationship between specific medications for these risk factors and the reduction of lacunar infarction risk is still under investigation. This is an area where further research is needed. 

Recently, Matthew Traylor et al. made substantial progress in identifying the genetic mechanisms underlying lacunar stroke by genome-wide association studies (GWAS) [2]. Their research has shed light on novel mechanisms underlying lacunar stroke pathogenesis, pointing to pathways that could potentially be targeted by more precision therapeutics [3].

The Mendelian randomization (MR) approach is a popular genetic epidemiological method that uses genetic variants as instrumental variables (*IVs*) for exposure to assess causal associations between risk factors and disease [4]. This method exploits the random allocation of genetic variants at conception to minimize any bias due to confounding and reverse causation that can restrict causal inference in observational research [5]. Genetic variation in drug-target proteins, such as *HMGCR*, can be leveraged to extend the application of Mendelian randomization (MR) to investigate drug effects [5,6]. Specifically, single-nucleotide polymorphisms (SNPs) in or near the *HMGCR* gene were used as proxies for *HMGCR* inhibition by statins [7]. Drug repurposing, otherwise known as drug repositioning, is a strategy that seeks to identify new indications and targets for approved drugs that are beyond the scope of their original medical indications [8]. Drug repurposing can be a time- and cost-effective way to discover novel therapeutics. Therefore, drug repurposing can be a useful strategy for lacunar stroke by exploring the potential of existing drugs that have already been proven safe and effective in humans [9]. 

Thus, to estimate which class of medication for antihypertensive, lipid-lowering, and antidiabetic drugs can exert a causal relationship with reducing the risk of lacunar infarction, we conducted this comprehensive MR analysis. We first exploited a two-sample MR approach to examine the causal associations of modifiable risk factors with lacunar stroke. Second, multivariable MR was conducted to estimate the direct causal effect of blood pressure and lipids on lacunar stroke. Third, drug-targeted MR was applied to evaluate several commonly used classes of antihypertensive, lipid-lowering agents and *GLP1R* agonism for hypoglycemic drugs likely to have efficacy in preventing lacunar stroke. The study design is presented in Figure 1. 

The purpose of this drug-target MR study was to determine whether some pre-existing drugs had a causal effect on lacunar stroke, with the ultimate goal of repurposing these drugs for new therapeutic applications. The study proposed that certain categories of medication could be identified and prioritized for use in the prevention of lacunar stroke.

## 2. Materials and Methods

We utilize a method known as Mendelian randomization (MR), which uses genetic variants as instrumental variables to estimate the causal effect of an exposure on an outcome. Our data sources include published genome-wide association studies (GWASs) and summary data from the MR base. Through this rigorous approach, we aim to shed light on the complex interplay of genetic and modifiable risk factors in the development of lacunar stroke, ultimately contributing to improved prevention and treatment strategies.

### 2.1. Potential Risk Factors

We considered potential risk factors that can be grouped under the following categories: anthropometry (waist-to-hip ratio, body fat, height, body mass index, bone mineral density, childhood BMI, birth weight), socioeconomic (education, intelligence), lifestyle/dietary (diphenylamine, smoking (number), eicosapntemacnioc acid, linoleic acid (LA; 18:2,n6), coffee, morning person, subjective well-being, adrenic acid (22:4,n6), arachidonic acid (AA; 20:4,n6), sedentary, carbohydrate, gamma linolenic acid (GLA; 18:3,n6)), cardiometabolic (common carotid intima-media thickness, coronary heart disease, HDL cholesterol || id:ieu-b-109, total cholesterol, homocysteine (Hcy), C-reaction protein (CRP), type 2 diabetes, lipoprotein(a), fasting glucose, heart rate, fasting proinsulin, fasting insulin, pulse pressure, apolipoprotein A-I || id:ieu-b-107, diastolic blood pressure (DBP), triglycerides || id:ieu-b-111, hypertension, fibrinogen, adiponectin, atrial fibrillation, leptin), LDL cholesterol || id:ieu-b-110, *HbA1C*, 2h glucose, systolic blood pressure (SBP), apolipoprotein B || id: ieu-b-108), endogenous substances (serum creatinine, vitamin E, uric acid, eGFRcrea, blood urea nitrogen, vitamin b12, protein, vitamin D), neuropsychiatric disorders (anorexia nervosa, schizophrenia, neuroticism, major depression, Parkinson’s disease) and other system diseases (chronic obstructive pulmonary disease, rheumatoid arthritis, osteoporotic fracture, Crohn’s disease, asthma). These 65 risk factors are listed in Appendix A. The eligible risk factors had the most solid evidence from previous observational studies, indicating that they may predispose an individual to lacunar stroke.

### 2.2. Data Sources

We searched PubMed for published genome-wide association studies to obtain the summary data (effect size estimates and standard errors) of risk factors. We also derived data from the MR base https://gwas.mrcieu.ac.uk/ (accessed on 29 August 2023) [10]. Details on the risk factors (including traits, number of IVs in the study (*p* < 5 × 10^−8^), the GWASs that the traits were interested, number of samples that the GWASs included, and units of the traits) that showed significant effects on lacunar stroke from which we obtained summary data for the current analyses are presented in Appendix A.

Lacunar stroke data were obtained from meta-analyses conducted in Europe, the USA, and Australia [2]. The meta-analyses included previous genome-wide association studies (GWASs) and additional cases and controls from the U.K. DNA lacunar stroke studies and the International Stroke Genetics Consortium. The study comprised a total of 6030 cases and 248,929 controls of European ancestry. In addition, the transethnic analysis included 7338 cases and 254,798 controls [11,12,13,14]. These lacunar stroke cases were MRI-confirmed cases, as MRI confirmation of lacunar stroke is more reliable than standard phenotyping. Genotyping arrays, quality control filters, and imputation reference panels could be found in the study [2]. Summary-level GWAS data could be derived from https://cd.hugeamp.org/downloads.html (accessed on 23 August 2023).

### 2.3. Genetic Variants

For univariable MR analyses, SNPs with genome-wide significance (*p* < 5 × 10^−8^) for each risk factor and their effect size estimates and standard errors were also collected. Only independent variants are not in linkage disequilibrium (defined as r^2^ < 0.001) with other genetic variants for the same risk factor. The IVs (F statistic > 10) for all the exposures were sufficiently informative [15]. F-statistics were calculated for each variant using the formula F = beta^2^/SE^2^ [16].

For blood pressure (BP)-related (SBP: systolic blood pressure; DBP: diastolic blood pressure; PP: pulse pressure) or lipid-related (ApoB: apolipoprotein B; LDL-C: low-density lipoprotein; TG: triglyceride; ApoA1: apolipoprotein A-I; HDL-C: high-density lipoprotein) traits, one exposure is genetically correlated with other exposures. Thus, we employed multivariable MR (MVMR) analysis to estimate the independent causal effect of each exposure [17,18]. As an extension of univariable MR, MVMR concatenating a set of IVs for each exposure estimates the direct effect of exposures on lacunar stroke risk, whereas univariable MR calculates the total effect. SNPs for the BP traits were obtained from summary statistics of a large GWAS of BP traits with over 1 million people of European ancestry [19]. All SNPs with genome-wide significance (*n* = 255, LD_r^2^ < 0.1) that were associated with any of the BP traits were included in the set of IVs. For the lipid multivariable MR analyses, in Model 1, we pooled all SNPs with genome-wide significance that were associated with any of the traits, including ApoB, LDL-C, and TG. In Model 2, the traits included ApoA1 and HDL-C [20]. The IVs were derived from the MR base with a threshold clump_r^2^ = 0.001 and clump_kb = 1000. In all, 384 and 435 IVs were concatenated for Model 1 and Model 2.

For antihypertensive drug-MR, MR analyses were performed to estimate the effect of a 10 mmHg reduction in blood pressure by antihypertensive drugs. Genetic instruments were selected based on their association with each blood pressure (BP) trait at genome-wide significance (*p* < 5 × 10^−8^) and their proximity to genes (near (+/−200 kb) or within encoding protein targets of 12 antihypertensive medication classes. Effect estimates for each genetic variant were derived for each BP trait from the trans-ancestry BP GWAS [21,22,23] (Appendix A). The primary analysis focused on the SBP-lowering effect, with sensitivity analyses considering the remaining BP traits (DBP, PP).

To investigate the effects of lipid-lowering drugs, the Mendelian randomization (MR) approach used *HMG*-CoA reductase as a proxy for statins. Five single-nucleotide polymorphisms (SNPs) associated with low-density lipoprotein (LDL) cholesterol at the genome-wide significant level (*p*  <  5.0  ×  10^−8^) and located within ±100 kb windows from the gene region of HMGCR were obtained (Figure 2). Variants located in the HMGCR, PCSK9, and NPC1L1 regions were selected using the method described by B. A. Ference [24]. The method proposed by Do et al. was used to select variants located in or near the APOC3 regions [25]. Variants were selected based on their associations with either low-density lipoprotein cholesterol (LDL-C) or triglycerides (TG), and they were not highly correlated (r^2^ < 0.4 or r^2^ < 0.3) (Appendix A).

For antidiabetic drugs, genetic proxies for glucagon-like peptide 1 receptor (GLP1R) agonism and glycemic control by any mechanism estimated to be associated with glycated hemoglobin (mmol/mol) were derived from 337,000 samples in the U.K. Biobank [26,27]. The linkage disequilibrium *r*^2^ values for variants used as proxies for GLP1R agonism and glycemic control were *r*^2^ < 0.1 and 0.001, respectively (Appendix A).

### 2.4. Mendelian Randomization Analysis

The principle and main analyses were described in our previous study [20,28,29]. Inverse-variance weighted (*IVW*) was the primary MR approach in the study. MR–Egger [30,31], the weighted median [32], and the simple median were calculated, and the MR–Egger intercept test was used to assess horizontal pleiotropy. We also used the Cochran Q statistic to test for heterogeneity and pleiotropy [33]. For instruments with only 1 variant, Wald-ratio MR was performed.

Next, the multivariable IVW method was used as the primary approach in conducting multivariable MR [33,34,35]. Univariable MR results reflect the total effect of each exposure on the outcome, including both direct and indirect effects through interactions with other exposures. Multivariable MR is often used in lipid analysis to estimate the direct causal effect of each exposure on an outcome [17]. Mendelian randomization (MR) can also provide valuable information about drugs, such as predicting their efficacy and revealing target-mediated adverse effects, which is also known as drug-target MR [5]. Drug-target MR can demonstrate the effect of modifying biomarkers through specific therapeutic targets on long-term health outcomes [36]. To account for both measured and unmeasured pleiotropy, we also used the multivariable MR-Egger [34] and the MR-Lasso method [37]. To test for heterogeneity and pleiotropy, we performed the Cochran Q statistic and multivariable MR-Egger test (intercept) [17,37].

The analyses were performed with R version 4.1.1 (R Core Team, Vienna, Austria) and the “Two Sample MR” (version 0.5.6) and “Mendelian Randomization” (version 0.5.1) packages [38]. Given that there was only one outcome under investigation (lacunar stroke), we used a 2-tailed *p*-value < 0.05 to denote evidence against the null hypothesis (i.e., *p* < 0.05 provided evidence in favor of an association between the exposure and outcome). 

## 3. Results

Among all 65 risk factors (i.e., anthropometric, serum substances, socioeconomic, lifestyle/dietary, cardiometabolic, and inflammatory factors), not surprisingly, genetically predicted hypertension, hyperlipidemia, and type 2 diabetes were identified as the predominant high-risk factors for the development of lacunar stroke (Figure 3). Greater height, higher educational level, fibrinogen, and atrial fibrillation have also been found to be associated with the incidence of lacunar stroke (Figure 3). 

The odds ratio for lacunar stroke estimated for a 1-SD increase in predisposition to elevated SBP was 1.06, and the effect was also validated and similar with SBP through analysis estimated for DBP and PP. 

For lipids, genetically predicted 1-SD increases in triglyceride and apolipoprotein B levels showed a causal detrimental effect on lacunar stroke, respectively. 

The analysis showed a decreased risk of lacunar stroke with genetically predicted high levels of apolipoprotein A-I and HDL. A genetic predisposition to type 2 diabetes significantly increases the risk of lacunar stroke. In addition, a high level of fasting proinsulin had a detrimental influence on the risk of lacunar stroke. The main results of significant risk factors in univariable MR are presented in Appendix A.

In blood pressure MVMR analysis, we found little evidence for the direct effects of any blood pressure factor on the risk of lacunar stroke. Specifically, the ORs of lacunar stroke per 1-SD increase in SBP, DBP, and PP were 0.97, 1.08, and 1.07, respectively. For lipid MVMR, when ApoB, LDL-C, and TG were assessed together in Model 1 using the multivariable *IVW* method, elevated TG levels remained significantly associated with a higher risk of lacunar stroke. In Model 2, neither apolipoprotein A-I nor HDL levels showed direct effects on lacunar stroke risk. The MVMR results are listed in Appendix A, and the MR–Egger intercept and Q test results in the study are listed in Appendix A.

Furthermore, the antihypertensive drug MR demonstrated that genetic variants mimicking the effect of calcium channel blockers (CCBs) showed the most potent effects in preventing lacunar stroke for each 1 SD decrease in genetically predicted SBP. The protective effects remained stable in the sensitivity analyses when variants mimicking CCB effects were estimated using DBP and PP (Figure 4). We also found that loop diuretics may prevent lacunar stroke, as estimated by SBP and DBP (Figure 4). Genetically predicted alpha-adrenoceptor blockers may have marginally protective effects on lacunar stroke, but only when estimated with SBP (Figure 4). No evidence of efficacy was identified for other antihypertensive drug classes in the analysis.

In the LDL-lowering target MR, the LDL-lowering effect predicted by the genetic variants at or near the *HMGCR* gene (i.e., mimicking the effect of statins), *NPC1L1* (mimicking the effects of ezetimibe), and *APOC3* (mimicking antisense anti-apoC3 agents) may decrease the risk of lacunar stroke (Figure 5). Genetic proxies for *HMGCR* agonism were associated with a reduced risk of lacunar stroke, and genetic proxies for *NPC1L1* inhibition were associated with a reduced risk of lacunar stroke. Significant protective associations of genetically proxied *APOC3* inhibition estimated either by LDL or triglyceride lowering with lacunar stroke risk were also observed (Figure 5).

We found little evidence for genetic proxies for *GLP1R* agonism effects on lacunar stroke; genetically proxied *GLP1R* agonism showed a marginal effect on lacunar stroke, while a genetically proxied improvement in overall glycemic control was associated with a reduced risk of lacunar stroke.

There were causal associations between genetically predicted greater height, higher education level, and lower odds of lacunar stroke, respectively. We also found that a high level of fibrinogen was associated with an increased risk of lacunar stroke, and a high risk of atrial fibrillation was associated with a decreased risk of lacunar stroke. No other causal evidence of lacunar stroke risk was found in the analysis. The main results of significant risk factors in univariable MR are presented in Appendix A.

Our MR analysis provided strong genetic evidence that hypertension, hyperlipidemia, and type 2 diabetes were the predominant risk factors for the development of lacunar stroke. Moreover, our MVMR analysis documented that genetically predicted elevated TG levels were still associated with a higher risk of lacunar stroke. Importantly, the comprehensive drug-target MR approach identified the protective effects of common antihypertensive, lipid-lowering medications on lacunar stroke. CCBs, statins, ezetimibe, and anti-*apoC3* agents were most likely to have potential effects of preventing lacunar stroke. *GLP1R* agonism did not show such effects, but improvement in overall glycemic control was associated with a reduced risk of lacunar stroke.

## 4. Discussion

In conclusion, this study provides valuable insights into the genetic and modifiable risk factors for lacunar stroke. The findings could potentially guide the development of preventive strategies and treatments for this condition [39]. This study indicated that old drugs could be repurposed to prevent lacunar stroke more precisely with substantially lower overall development costs and shorter development timelines [40]. Such drugs should be given high priority when doctors are making decisions and may contribute to a healthier brain during aging.

Antihypertensive, lipid-lowering agents have been explored with stroke in a previous study. Georgakis MK et al. found that genetic proxies for CCBs showed an inverse association with the risk of ischemic stroke compared with proxies for β-blockade, which was particularly strong for small vessel stroke [41]. Our study showed that among the 12 antihypertensive drug classes, CCBs had the most potent effects in preventing lacunar stroke and that loop diuretics may prevent the development of lacunar stroke, as estimated by SBP. β-Blockade did not show protective effects against the risk of lacunar stroke estimated by SBP. Large-scale meta-analyses of clinical trials have shown that calcium channel blockers have a stronger effect on reducing the risk of stroke than β blockers [42,43]. However, the association between lipid-lowering variants mimicking statin use and a lower risk of ischemic stroke was statistically significant only for large artery stroke [44]. The possible explanation was that the data we used as outcomes made substantial progress in identifying the genetic mechanisms underlying lacunar stroke. We also found little evidence for the effect of *PCSK9* inhibitors in preventing lacunar stroke. Our findings agree with the results by Hopewell JC et al. in that *PCSK9* inhibitors are unlikely to have an effect on lacunar stroke risk. Notably, genetic proxies for *HMGCR* agonism, *NPC1L1* (mimicking the effects of ezetimibe) and *APOC3*, were predicted to decrease the risk of lacunar stroke; they do affect the risk of lacunar stroke, but they may not do so via LDL. We found little evidence to suggest that LDL itself affects the risk of developing lacunar stroke. The underlying mechanism needs to be explored in the future.

Blood pressure showed a significant association with lacunar stroke risk in univariable MR but not in MVMR. A possible explanation for this is that MVMR analysis is used to estimate the independent causal effect of each exposure, whereas univariable MR is used to calculate the total effect. Elevated TG levels were found to be associated with a higher risk of lacunar stroke conditional on ApoB and LDL-C levels in multivariable MR, which was consistent with the results from univariable MR. This result suggests that compared with ApoB or LDL-C, TG may be more likely to have a detrimental causal effect on lacunar stroke. Thus, in drug-targeted MR, we applied proxies for lipid-lowering agents estimated by TG-lowering targets.

In this screening Mendelian randomization analysis, we also found that higher height and education levels are associated with a reduced risk of lacunar stroke. Currently, it is well understood that higher education levels often indicate that individuals are more likely to have access to healthy diets, appropriate physical exercise, and harmless work environments and are more concerned about their health status. These characteristics are all favorable factors in reducing the risk of lacunar stroke. However, the biological mechanism behind the association between higher height and reduced risk of lacunar stroke is currently unclear. For both of these results, further mediation analysis would be helpful in uncovering their underlying biological mechanisms. 

The relationship between fibrinogen and lacunar stroke can be explained by the fact that high levels of fibrinogen can lead to the formation of blood clots, which can block small arteries in the brain and cause a lacunar stroke [45]. Further research is needed to understand the underlying biological mechanisms behind this association. However, this finding suggests that monitoring fibrinogen levels may be an important factor in reducing the risk of lacunar stroke [46].

Additionally, this result may imply that atrial fibrillation patients are less likely to develop lacunar stroke. However, more research is needed to confirm this hypothesis and to determine the underlying biological mechanisms. One possible explanation for this association is that atrial fibrillation may lead to changes in blood flow and pressure within the brain, which could reduce the risk of lacunar stroke. Another possibility is that the medications used to treat atrial fibrillation may also have a protective effect against lacunar stroke. Future studies could investigate the effects of different types of medications on the risk of lacunar stroke in individuals with atrial fibrillation, as well as the potential role of lifestyle factors such as diet and exercise.

The term lipohyalinosis refers to a concentric accumulation of hyaline material in the walls of small cerebral vessels, which causes narrowing and blockage of the penetrating arteries [47]. This is among the first and most frequent mechanisms of lacunar stroke that have been documented and confirmed by pathology. High blood pressure may cause the vessel walls to thicken and degenerate, as well as foam cells to fill up the lumen of small arteries that penetrate the brain, leading to lipohyalinosis. Diabetes is a condition that affects the metabolism of blood glucose (or blood sugar) and blood lipids, causing chronic inflammation [48]. These factors harm the vessel wall, leading to the accumulation of lipid, fibrous tissue, and calcification and the formation of atherosclerotic plaques. Our study suggests that CCBs, statins, ezetimibe, and anti-*apoC3* agents may prevent lacunar stroke by lowering blood pressure and lipid levels. Additionally, the potential role of statins in improving lacunar stroke might be due to the anti-inflammatory effect of statins [49]. The antihyperglycemic effect alone is able to reduce oxidative stress, with improvement in endothelial function, which is one of the triggers for the Virchow triad [50]. Future studies should investigate whether other mechanisms are also involved, which may help identify new targets for interventions or therapies. A strength of our study was the use of two-sample MR, which allowed us to utilize the latest GWASs for lacunar stroke outcomes, which included 7338 cases and 254,798 controls [2]. MR is a more effective approach than traditional pharmacoepidemiological methods in addressing certain types of confounding. This includes confounding by indication, as well as confounding by environmental and lifestyle factors that cannot be fully adjusted for using observational data. Residual confounding is an inevitable consequence of measurement error and incomplete capture of all potential confounding factors. Importantly, drug repurposing, being a less expensive and time-consuming approach, brings effective therapies to patients compared with the cumbersome traditional processes of discovery and development. Repurposing candidates have already undergone several stages of clinical development and have well-established safety and pharmacological profiles. This translates to lower development costs, faster development times, and ultimately lower out-of-pocket costs for patients, reducing the actual cost of therapy [51].

Several limitations merit consideration. Our study was constrained by the fact that MR estimates the effect of lifelong exposure, while drugs typically have much shorter periods of exposure [52]. Additionally, systolic blood pressure may have age-dependent effects. As a result, the effect sizes we estimated may not directly reflect what is observed in trials or clinical practice and may not be able to identify critical periods of exposure. Nevertheless, our study assumed a linear relationship between exposures and lacunar stroke and did not investigate the nonlinear effects of the exposures [30,32]. The populations of exposures and outcomes we explored in the study were not all from subjects of European ancestry; the lacunar stroke GWAS data were derived from transethnic studies, and the underlying populations were primarily composed of individuals of European ancestry [53]. Thus, bias from population stratification is deemed likely [54]. In this analysis, we only analyzed *GLP1R* agonism for hypoglycemic drugs and did not analyze other commonly used hypoglycemic drugs in clinical practice, such as metformin and *DPP4* inhibitors, which may result in missing information. Metformin is a multi-target drug that is not suitable for MR analysis [55], and there are not enough available instrumental variables to proxy DPP4 inhibitors as far as we know. Finally, completely ruling out pleiotropy or an alternative direct causal pathway is a challenge for all MR analyses because there are probably some unknown confounders that could influence lacunar stroke [56]. These limitations highlight the importance of cautious interpretation of findings [54]. Further research is needed to validate these findings and to explore their clinical implications.

Precision medicine technology is continuously developing, and we believe that it will play an increasingly important role in the diagnosis and treatment of lacunar stroke [57]. To foster the potential of MR analysis, it is crucial to acquire large datasets that comprise subject-level information on hundreds to thousands of patients [58]. This will enable the development of more accurate and reliable predictive models for lacunar stroke, which can help clinicians identify high-risk patients and provide them with timely and effective interventions [59]. In conclusion, precision medicine, particularly using drug-target MR, is a promising approach for the prevention and treatment of lacunar stroke and an area that deserves further research and development [9,39,60].

## 5. Conclusions

Using an MR design to comprehensively repurpose approved drugs to precisely prevent lacunar stroke with well-established safety and low costs. CCBs, statins, ezetimibe, and anti-*apoC3* agents were most likely to have potential effects of preventing lacunar stroke. This study provided solid evidence for doctors to consider when making decisions in clinical practice and may contribute to a healthier brain during aging.

## Figures and Tables

**Figure 1 biomedicines-12-00017-f001:**
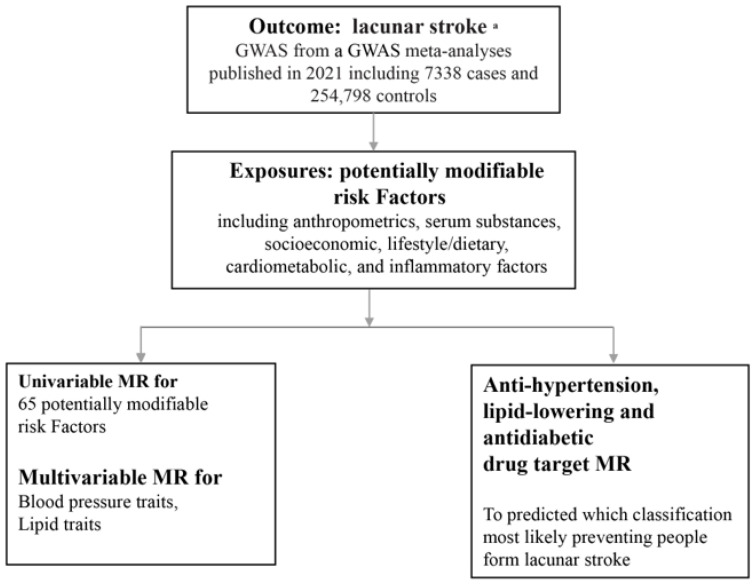
Overall study design. ^a^: [2]. GWAS: genome-wide association study; MR: Mendelian randomization.

**Figure 2 biomedicines-12-00017-f002:**
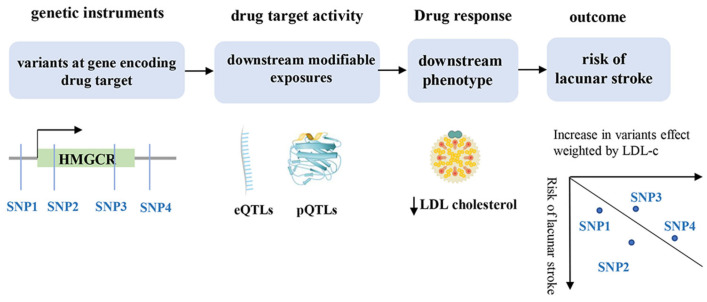
Principles of drug-target MR analysis framework. MR: Mendelian randomization. SNP: single-nucleotide polymorphism; HMGCR: 3-Hydroxy-3-Methylglutaryl-CoA reductase; eQTLs: expression quantitative trait loci; pQTLs: protein quantitative trait loci; LDL: low-density lipoprotein.

**Figure 3 biomedicines-12-00017-f003:**
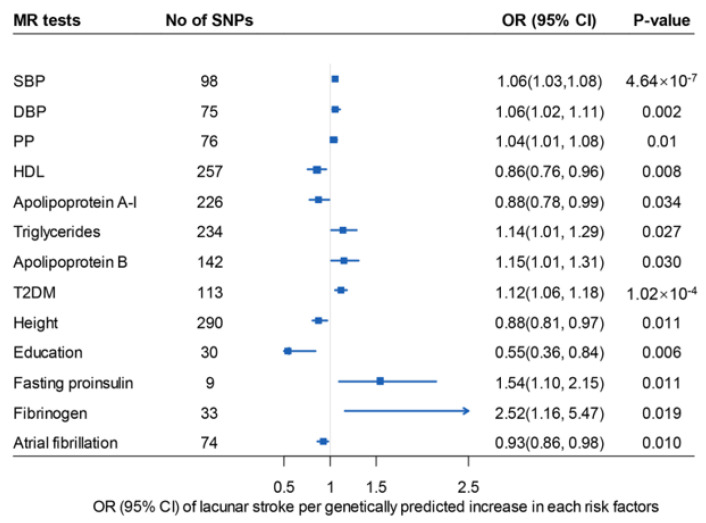
Odds ratios for associations between genetically predicted significant factors and lacunar stroke conduct by univariable MR. MR: mendelian randomization. DBP: diastolic blood pressure; PP: pulse pressure; SBP: systolic blood pressure; HDL: high-density lipoprotein; T2DM: diabetes mellitus type 2; OR: odds ratio. 95% CI: 95% confidence interval. HDL, apolipoprotein A-I, triglycerides, apolipoprotein B, height used the multiplicative random effects model due to instrumental heterogeneity (Cochran Q test *p* < 0.05).

**Figure 4 biomedicines-12-00017-f004:**
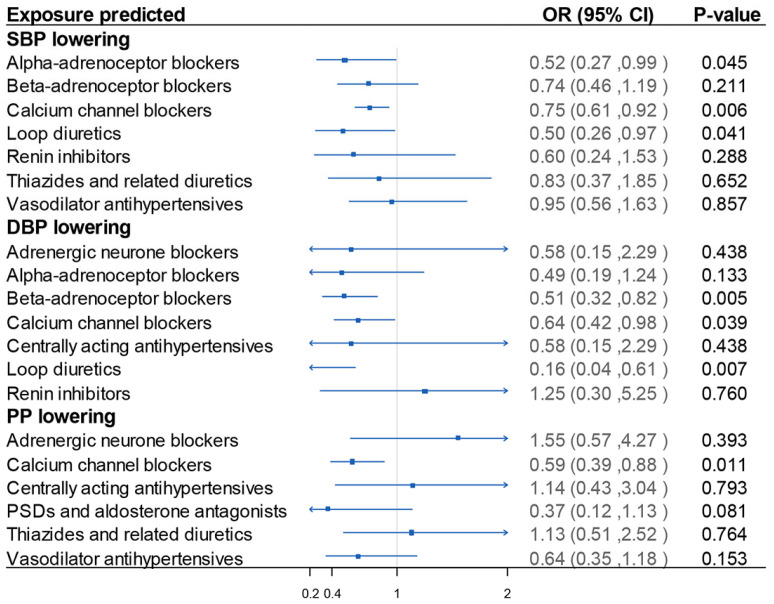
Results of blood pressure-lowering drug-target MR analysis. Estimated for SBP-lowering target weighted by SBP, DBP-lowering target weighted by DBP, and PP-lowering target weighted by PP. On the left of *x*-axis = 1 presented drug use protective, and on the right of *x*-axis = 1 presented drug use detrimental. MR: Mendelian randomization; DBP: diastolic blood pressure; PP: pulse pressure; SBP: systolic blood pressure. OR: odds ratio. 95% CI: 95% confidence interval.

**Figure 5 biomedicines-12-00017-f005:**
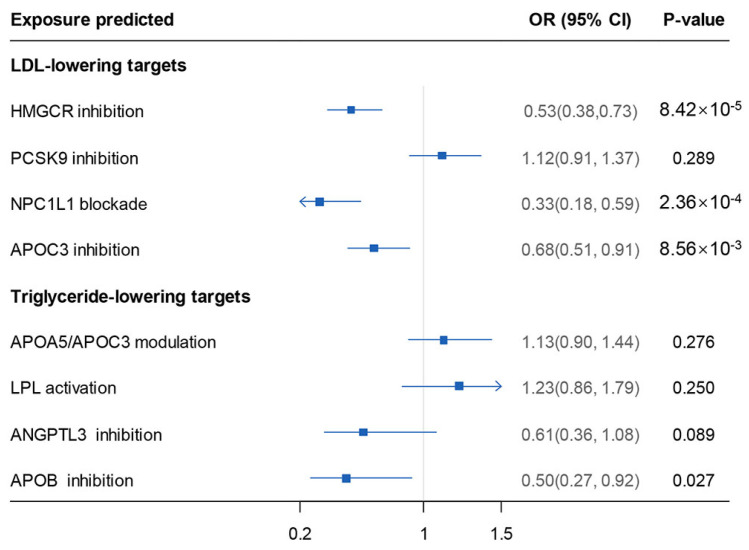
Results of lipid-lowering drug-target MR analysis. Estimated for LDL-lowering target weighted by LDL, triglyceride-lowering target weighted by triglyceride. On the left of *x*-axis = 1 presented drug use protective, and on the right of *x*-axis = 1 presented drug use detrimental. MR: Mendelian randomization; LDL: low-density lipoprotein; *HMGCR*: 3-Hydroxy-3-Methylglutaryl-CoA reductase; *PCSK9*: proprotein convertase subtilisin kexin type 9; *NPC1L1*: *NPC1*-like intracellular cholesterol transporter 1; *APOC3*: apolipoprotein C3; *APOA5*: apolipoprotein A5; LPL: lipoprotein lipase; *ANGPTL3*: angiopoietin-like 3; APOB: apolipoprotein B; OR: odds ratio. 95% CI: 95% confidence interval.

## Data Availability

All data collected for the study and code used in the analysis will be made available to others.

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
