# Peer review of "Unlocking the Medicinal Mysteries: Preventing Lacunar Stroke with Drug Repurposing"

_biomedicines, 2023, doi:10.3390/biomedicines12010017_

Round 1
Reviewer 1 Report
Comments and Suggestions for Authors
The article is very interesting and deals with a topic that has already been a source of discussion in the past. I believe that the study inevitably has some limitations, but all of them are well considered in the "limitations of the study" section.
Lines 90-92 “We considered potentially modifiable risk factors that can be grouped under the following categories: anthropometry ( waist-to-hip ratio, body fat, height, body mass index, bone mineral density, childhood BMI, birth weight), socioeconomic (education, intelligence), lifestyle/dietary…”
Some of the risk factors listed are not modifiable, for example height, intelligence and others are are only theoretically modifiable and in any case no longer modifiable at the time of observation (e.g. birth weight)
Lines 398-401 “we only analyzed GLP1R agonism for hypoglycemic drugs and did not analyze other commonly used hypoglycemic drugs in clinical practice, such as metformin and DPP4 inhibitors, which may result in missing information.” Since only GLP1RA were analyzed among the hypoglycemic agents, as written in lines 81-83 “Third, drug-targeted MR was applied to evaluate several commonly used classes of antihypertensive, lipid-lowering and antidiabetic agents likely to have efficacy in preventing lacunar stroke” should be revised.
Author Response
Dear Editor,
Manuscript, biomedicines-2702237
Repurposing antihypertensive, lipid-lowering and antidiabetic drugs for lacunar stroke
Thank you for your comments on the above manuscript and for the opportunity to revise it. We have taken the editor and reviewers’ comments and suggestions into careful consideration and revised the manuscript accordingly.
Response to editor comments:
1)We had rephrased the abstract into a single paragraph and broken down the introduction into smaller paragraphs for easier readability;
2) We had provided a brief overview of the paper's structure before Section Introduction.
3) In the section of Discussion, we had added a paragraph that included the several potential mechanisms of lacunar stroke, provided by which the risk factors exert their influence, thus give an explanation why some drugs could help in preventing lacunar stroke. This makes the article more in-depth and logical. (page 11, line 394-409)
“The term lipohyalinosis refers to a concentric accumulation of hyaline material in the walls of small cerebral vessels, which causes narrowing and blockage of the penetrating arteries. This is among the first and most frequent mechanisms of lacunar stroke that have been documented and confirmed by pathology. High blood pressure may cause the vessel walls to thicken and degenerate, as well as foam cells to fill up the lumen of small arteries that penetrate the brain, leading to lipohyalinosis. Diabetes is a condition that affects the metabolism of blood glucose (or blood sugar) and blood lipid, causing chronic inflammation. These factors harm the vessel wall, leading to the accumulation of lipid, fibrous tissue and calcification, and the formation of atherosclerotic plaques. Our study suggests that CCBs, statins, ezetimibe, and anti-apoC3 agents may prevent lacunar stroke by lowering blood pressure and lipid levels. Additionally, potential role of statins in im-proving lacunar stroke might be due to the antinflammatory effect of statins [30]. The antihyperglycemic effect alone is able to reduce oxidative stress, with improvement in endothelial function, which is one of the triggers for Virchow triad [31]. Future studies should investigate whether other mechanisms are also involved, which may help identify new targets for interventions or therapies.”
4) we had added the purpose and hypotheses that the paper aims to address at the end of section of Introduction. This highlights the purpose and the main points of the article. (page 3, line 108-111)
“The purpose of this drug-target MR study was to determine whether some pre-existing drugs had a causal effect on lacunar stroke, with the ultimate goal of re-purposing these drugs for new therapeutic applications. We propose that certain categories of medication could be identified and prioritized for use in the prevention of lacunar stroke.”
5) We reported the effect sizes and statistical significance of the relevant associations in both the abstract and the main text.
6) We had revised the duplicated sentence together during the revision process, which was also tracked.
In the revised paper, please find our point-by-point responses to the reviewers’ concerns in the order that they were originally listed, and details of the pages on which the changes have been made.
The manuscript has not been submitted nor is it under consideration for publication by another journal. All authors have read the manuscript and are in agreement that the work is ready for submission and accept the responsibility for manuscript contents. None of the authors have any conflict of interest in the matter.
We believe that the quality of the manuscript has been considerably enhanced as a consequence of the review process. We hope that the revised paper now meets your approval for publication in Biomedicines. Please do not hesitate to contact me if you need any further information.
Sincerely,
Dongsheng Fan
Department of Neurology,
Peking University Third Hospital,
49 North Garden Road, Haidian District, Beijing 100191, People’s Republic of China
Phone: (+86)13701023871
Fax: 086-010-82266250
Email: dsfan2010@aliyun.com
Reviewer 1:
Thank you for your comments on the above manuscript and for the opportunity to revise it.
1 We have changed the phrase into “Potential risk factors”.
2 We have changed the sentence into “Third, drug-targeted MR was applied to evaluate several commonly used classes of antihypertensive, lipid-lowering and GLP1R agonism for hypoglycemic drugs likely to have efficacy in preventing lacunar stroke.”
Reviewer 2:
Thank you for your comments on the above manuscript and for the opportunity to revise it.
we had added the purpose and hypotheses that the paper aims to address at the end of section of Introduction. This highlights the purpose and the novelty of the article.
“The purpose of this drug-target MR study was to determine whether some pre-existing drugs had a causal effect on lacunar stroke, with the ultimate goal of re-purposing these drugs for new therapeutic applications. We propose that certain categories of medication could be identified and prioritized for use in the prevention of lacunar stroke.”
“Repurposing several drugs with well-established safety and low costs for lacunar stroke prevention should be given high priority when doctors are making decisions in clinical practice and may con-tribute to healthier brain aging.”
Reviewer 2 Report
Comments and Suggestions for Authors
I have carefully read the presented manuscript and in my opinion it concerns very interesting and up-to-date theme. The title of the manuscript is very encouraging and indicates that we will be dealing with completely different drugs than those previously used in the treatment of stroke. Unfortunately, the content of the manuscript is very disappointing. There is no information here about repositioned drugs, but rather about those that have been used for a long time in the prevention and treatment of stroke. Additionally, I lack the exact aim of the presented manuscript, and the Authors themselves repeat the phrases many times:
"this finding and consistent…."
"supports..."
“similar analyzes have also been performed for….”
what clearly indicates that the information presented in the manuscript is already widely known. Therefore, in my opinion, there is no novelty in the manuscript.
Author Response
Reviewer 2:
Thank you for your comments on the above manuscript and for the opportunity to revise it.
we had added the purpose and hypotheses that the paper aims to address at the end of section of Introduction. This highlights the purpose and the novelty of the article.
“The purpose of this drug-target MR study was to determine whether some pre-existing drugs had a causal effect on lacunar stroke, with the ultimate goal of re-purposing these drugs for new therapeutic applications. We propose that certain categories of medication could be identified and prioritized for use in the prevention of lacunar stroke.”
“Repurposing several drugs with well-established safety and low costs for lacunar stroke prevention should be given high priority when doctors are making decisions in clinical practice and may con-tribute to healthier brain aging.”
Reviewer 3 Report
Comments and Suggestions for Authors
I read with great interest the article titled “Repurposing Antihypertensive, lipid-lowering and Antidiabetic Drugs for the Prevention of Lacunar Stroke” by Linjing Zhang et al.
The paper's design is sound, and the article is logically organized into appropriate sections and subsections. English is fine, only minor spell check.
Here is the comment and suggested revision:
1. Title: by reading the title it was not clear to me the type of study you wanted to performe. Please modify.
2. Abstract: MR please report it in extended form the first time.
3. Discussion: The authors have pointed out 3 main aspects to prevent lacunar stroke. However, nothing has been discussed about antidiabetic medications. In particular, diabetic patients are at increased risk of developing lacunar stroke, dyslipidemia, atrial fibrillation… This subpopulation should be better discussed. In addition, several old studies have reported a well tolerability to statins in this population (doi: 10.1046/j.1463-1326.2000.00106.x) and the potential role of statins in improving lacunar stroke might be due to the antinflammatory effect of statins (doi: 10.1007/s11883-021-00977-6). For what regards antidiabetic medications, this improving could be driven by several aspects. Not taking into account newer drugs, the antihyperglycemic effect alone is able to reduce oxidative stress, with improvement in endothelial function (doi: 10.3390/cimb45080420), which is one of the triggers for Virchow triade. Please discuss further these aspects.
4. Please report why only GLP1RA was taking into account.
5. Please write a future perspective paragraph also highlighting the importance of precision medicine.
Comments on the Quality of English Language
English is fine
Author Response
1 we had changed the title into “Repurposing Antihypertensive, lipid-lowering and
Antidiabetic Drugs for the Prevention of Lacunar Stroke from a Mendelian randomization study”.
2 we had extended the “MR” into “Mendelian randomization”.
3 we had added a paragraph that included the several potential mechanisms of lacunar stroke, provided by which the risk factors exert their influence, thus give an explanation why some drugs could help in preventing lacunar stroke. This makes the article more in-depth and logical. (page 11, line 397-409)
“The term lipohyalinosis refers to a concentric accumulation of hyaline material in the walls of small cerebral vessels, which causes narrowing and blockage of the penetrating arteries. This is among the first and most frequent mechanisms of lacunar stroke that have been documented and confirmed by pathology. High blood pressure may cause the vessel walls to thicken and degenerate, as well as foam cells to fill up the lumen of small arteries that penetrate the brain, leading to lipohyalinosis. Diabetes is a condition that affects the metabolism of blood glucose (or blood sugar) and blood lipid, causing chronic inflammation. These factors harm the vessel wall, leading to the accumulation of lipid, fibrous tissue and calcification, and the formation of atherosclerotic plaques. Our study suggests that CCBs, statins, ezetimibe, and anti-apoC3 agents may prevent lacunar stroke by lowering blood pressure and lipid levels. Additionally, potential role of statins in im-proving lacunar stroke might be due to the antinflammatory effect of statins [30]. The antihyperglycemic effect alone is able to reduce oxidative stress, with improvement in endothelial function, which is one of the triggers for Virchow triad [31]. Future studies should investigate whether other mechanisms are also involved, which may help identify new targets for interventions or therapies.”
4 we had reported why only GLP1RA was taking into account. “While metformin is a multi-target drug which is not suitable for MR analysis, and there are not enough available instrumental variables to proxy DPP4 inhibitors as we know so far.”
5 we had written a future perspective paragraph also highlighting the importance of precision medicine.
“Precision medicine technology is continuously developing, and we believe that it will play an increasingly important role in the diagnosis and treatment of lucunar stroke [33]. To foster the potential of MR analysis, it is crucial to acquire large datasets that comprise subject-level information on hundreds to thousands of patients. This will enable the development of more accurate and reliable predictive models for lucunar stroke, which can help clinicians identify high-risk patients and provide them with timely and effective interventions. In conclusion, precision medicine, particularly using drug-target MR, is a promising approach for the prevention and treatment of lucunar stroke and an area that deserves further research and development [34].”
Round 2
Reviewer 2 Report
Comments and Suggestions for Authors
In my opinion this the manuscript has not been revised in a way that would enable its publication.
Author Response
Dear Editor,
Manuscript, biomedicines-2702237
Repurposing antihypertensive, lipid-lowering and antidiabetic drugs for lacunar stroke
Thank you for your comments on the above manuscript and for the second opportunity to revise it. We have taken the editor and reviewers’ comments and suggestions into careful consideration and revised the manuscript accordingly.
Response to editor comments:
- We had changed the title into “Unlocking the Medicinal Mysteries: Preventing Lacunar Stroke with Drug Repurposing”.
2)We had rephrased the abstract into a single paragraph and broken down the introduction into smaller paragraphs for easier readability; and the words count in this section was 219. The results were concluded with a single sentence that places the findings in a broader context. And no statistical details in the section.
3) we had listed ten keywords chosen from Medical Subject Headings (MeSH) according to this paper.
4) we had provided a graphical abstract.
5) Introduction: Basically, we had re-organized this section in a logical and cohesive manner. The word count was 610 and we think it should meet your expectation for this section. Now this paper we believe that it was easily understandable to a scientist of any discipline.
6) Methods: we had opened this section with a short introductory paragraph and citing more references to ensure the reliability and integrity of the evidence in the study design. “We utilize a method known as Mendelian Randomization (MR), which uses genetic variants as instrumental variables to estimate the causal effect of an exposure on an outcome. Our data sources include published GWASs and summary data from MR base. Through this rigorous approach, we aim to shed light on the complex interplay of genetic and modifiable risk factors in the development of lacunar stroke, ultimately contributing to improved prevention and treatment strategies.” We had cited more references to ensure the reliability and integrity of the evidence of the design.
7) Results: we had presented all the statistical values in the tables and avoid them in the text. We had closed this section with a paragraph that puts the results into a more general context.
“Our MR analysis provided strong genetic evidence that hypertension, hyperlipidemia and type 2 diabetes were the predominant risk factors for the development of lacunar stroke. Moreover, our MVMR analysis documented that genetically predicted elevated TG levels were still associated with a higher risk of lacunar stroke. Importantly, the comprehensive drug-target MR approach identified protective effects of common antihypertensive, lipid-lowering medications on lacunar stroke. CCBs, statins, ezetimibe, and anti-apoC3 agents were most likely to have potential effects of preventing lacunar stroke. GLP1R agonism did not show such effects, but improvement in overall glycemic control was associated with a reduced risk of lacunar stroke.
8)Discussion: we now present this section with 1507 words. we had opened this section with an introductory paragraph. “In conclusion, this study provides valuable insights into the genetic and modifiable risk factors for lacunar stroke. The findings could potentially guide the development of preventive strategies and treatments for this condition [39].” And according to your suggestion, we had re-construct this section and cited references to ensure the reliability and integrity of this section.
9) References: the number of references of this paper now was 60.
In the revised paper, please find our point-by-point responses to the reviewers’ concerns in the order that they were originally listed, and details of the pages on which the changes have been made.
We believe that the quality of the manuscript has been considerably enhanced as a consequence of the review process. We hope that the revised paper now meets your approval for publication in Biomedicines. Please do not hesitate to contact me if you need any further information.
Sincerely,
Dongsheng Fan
Department of Neurology,
Peking University Third Hospital,
49 North Garden Road, Haidian District, Beijing 100191, People’s Republic of China
Phone: (+86)13701023871
Fax: 086-010-82266250
Email: dsfan2010@aliyun.com
Reviewer 3 Report
Comments and Suggestions for Authors
The authors addressed all the issues I raised
Author Response

(The authors gave the same response as above.)
